# Gender bias in scholarly peer review

**Abstract** Peer review is the cornerstone of scholarly publishing and it is essential that peer reviewers are appointed on the basis of their expertise alone. However, it is difficult to check for any bias in the peer-review process because the identity of peer reviewers generally remains confidential. Here, using public information about the identities of 9000 editors and 43000 reviewers from the Frontiers series of journals, we show that women are underrepresented in the peer-review process, that editors of both genders operate with substantial same-gender preference (homophily), and that the mechanisms of this homophily are gender-dependent. We also show that homophily will persist even if numerical parity between genders is reached, highlighting the need for increased efforts to combat subtler forms of gender bias in scholarly publishing.

**MARKUS HELMER**[*]**, MANUEL SCHOTTDORF, ANDREAS NEEF AND DEMIAN BATTAGLIA**[*]

## Introduction

Peer review has an important role in improving the quality of research papers. It is the "life-blood of research in academia [. . .] the social structure that subjects research to the critical assessment of other researchers" (*Bourdieu, 1975*). This structure relies on self-regulated interactions within the scientific community, in which a journal editor appoints peer reviewers with expertise in the subject of a particular manuscript to report on the quality of that manuscript and to provide recommendations for its improvement. Other attributes of the peer reviewer, such as their gender, should be irrelevant (*Moss-Racusin et al., 2012*; *Nature, 2013*). However the identities of peer reviewers and editors are usually confidential, so previous work on gender balance in the peer-review process has relied on small, monodisciplinary data sets and these studies have given partly contradictory reports (*Lloyd, 1990*; *Gilbert et al., 1994*; *Budden et al., 2008*; *Borsuk et al., 2009*; *Knobloch-Westerwick et al., 2013*; *Larivière et al., 2013*; *Buckley et al., 2014*; *Demarest et al., 2014*; *Handley et al., 2015b*; *Fox et al., 2016*).

Frontiers journals (www.frontiersin.org) differ from most journals in that they generally disclose the identities of peer reviewers and associate editors alongside each published article in an attempt to increase the transparency and quality of the publication process (*Poynder, 2016*). This allowed us to extract the names of associate editors, peer reviewers and authors for articles published in Frontiers journals between 2007 (when the first Frontiers journal was published) and the end of 2015. This data set included the names of more than 9000 editors, 43,000 reviewers, and 126,000 authors for about 41,000 articles published in 142 journals in Science, Health, Engineering and the Humanities and Social Sciences (see Materials and methods). This data set is one of the largest available to date, and contains at least an order of magnitude more information than most data sets used in previous studies of peer review (see *Supplementary file 2* for comparison).

Analysis of this data set reveals that women are underrepresented in the peer-review process, and that editors of both genders operate with substantial same-gender preference (homophily) when appointing reviewers. Moreover, our analysis suggests that this homophilic tendency will persist even when men and women are fairly represented in the peer-review process. Our results confirm the need for increased efforts to

fight against subtler forms of gender bias in scholarly publishing and not just focus on numerical under-representation alone.

## Results

To assess whether our data set was representative of an active and mature research community, we created directed networks (*Figure 1a$_1$*), in which individual scientists appeared as vertices, while arrows denoted interactions between them ("*is appointing*" in the editor-to-reviewer network, and "*is editing (reviewing) a manuscript of*" in the editor (reviewer)-to-author network). As a whole, the networks had an exponentially fast growth in time, with a large fraction of people participating in a connected component of the graph reaching 90% of the total network size. Furthermore, graph theoretical metrics such as shortest path length, small-world index as well as several other network properties have changed little in the 3-5 last considered years (*Figure 1—figure supplement 1*). Thus, peer-reviewing interactions in the Frontiers journal series gave rise to a mature, topologically stable and integrated community, even though its contributors constitute only a small subset of researchers worldwide.

We then looked for signatures of gender bias and of its evolution across time in the structure of these large networks. We study first the fractions of assignments for reviewing or editing given to female or male scientists and, for comparison, we also show the fractions of author contributions. *Figure 1b* reveals that the fractions of authoring, reviewing and editing contributions by women — amounting to 37%, 28% and 26%, respectively, in the complete accumulated data until 2015 — are always significantly smaller than the corresponding fractions for men. The unbalance between male and female contributions thus worsens when gradually ascending through the peer-review hierarchy. Apart from a few outlier countries, this pattern was dominant worldwide (*Figure 1—figure supplement 2*). It was also largely present in all the considered journals when looking at them individually (*Figure 1c*). Overall, the number of contributions by female authors varies between about 15% (Frontiers in Neurorobotics) and 50% (Public Health), by female reviewers between about 15% (Surgery) and 50% (Public Health), and by female editors from ca. 5% (Robotics AI) to 35% (Aging Neuroscience). Globally, we observed a trend towards gender parity across time. The rates of change were, however, very slow. Linear extrapolation based on the fractions observed from 2012 to 2015 would predict that exact parity could be achieved as late as 2027 for authoring, 2034 for reviewing and 2042 for editing.

We wondered whether these lower fractions of contributions to the different roles were just due to the fact that overall there are numerically less female than male authors, reviewers and editors (39%, 30% and 28% out of all available authors, reviewers and editors, respectively, were women, closely mirroring the observed fractions of assignments). To test this hypothesis, we took the exact same network of peer-review interactions in the Frontiers journals for given, and randomly permuted gender labels among scientists of a given role (*Figure 1a$_2$*). This procedure maintained the ratio of female and male scientists acting in the different roles, but destroyed all direct correlations between gender and numbers of contributions. Repeatedly drawing random genders for the scientists in the network generated a surrogate ensemble that we used to estimate the expected number of contributions in a gender-blind control network. Author and reviewing contributions by women lay significantly below the confidence intervals obtained through this permutation testing procedure since 2009 and 2011, respectively. For female editing contributions we found the same, though non-significant, trend. Thus, the mere overall smaller number of female actors cannot explain the observed unbalanced fractions of female contributions to the peer-review chain.

We then looked for possible differences over the entire distributions of the number of peer-review tasks and authoring contributions for men and women. These distributions are fat-tailed (*Figure 2a-c*), indicating that some individuals provided a large number of contributions to the publication chain, while a majority of scientists authored, reviewed or edited only a small number of manuscripts. Moreover, comparing the observed degree distributions to the expectations derived from the same null hypothesis used above, women had a significantly smaller than chance probability to review (and author) more than one article, while their probability to act as single-time reviewer or author exceeded the expected chance level (*Figure 2e-f*). In the editing role, women underrepresentation was significant only for a high number of contributions. Furthermore, we found significant deviations from chance-level expectations across the

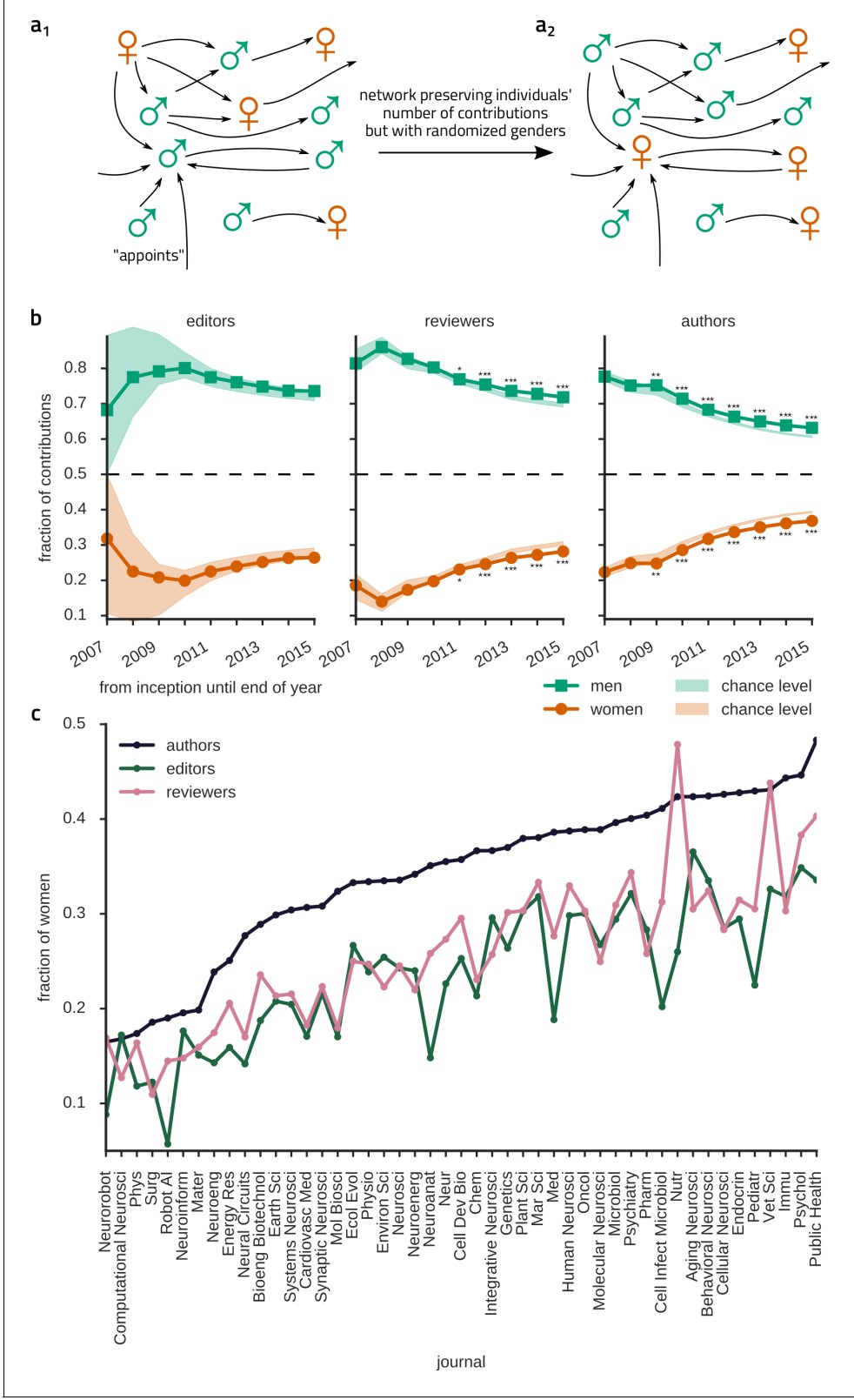

**Figure 1.** Women review and author even less articles than expected from their numeric underrepresentation. (a₁) We represent peer-reviewing interactions as directed graphs, in which vertices denote scientists. In the editor-to-reviewer network every edge represents the act of an editor (source vertex) appointing a reviewer (target vertex) to review a manuscript (and the reviewer has accepted the invitation). Analogously, in the reviewer-to-author network edges represent a reviewer reviewing a manuscript of an author. (b) The development of the fraction of contributions for each gender

*Figure 1 continued on next page*

Figure 1 continued

are shown for editors, reviewers and authors. Since the start of the Frontiers journals in 2007 until 2015, women (circles) edit, review and author much less than 50% of manuscripts, as expected from their numeric underrepresentation. However, the actual numbers of reviewing and authoring contributions by women are even smaller than expected by chance, taking into account their numeric underrepresentation. This is revealed by comparison with a null hypothesis in which gender and number of contributions are assumed to be independent. To this end, we generated surrogate ensembles by shuffling the genders of scientists appearing in a given role in the network ($a_2$). From the surrogate ensembles, we obtained 95% confidence intervals (CIs; shaded areas in b). *, **, *** over (under) the data symbols denote the data lying over (under) the 95%, 99%, 99.9% CIs. Note that for all three subnetworks, there is a noticeable, but extremely slow trend towards equity (dashed line) for the fraction of contributions. (c) The fraction of female contributors, ranked in increasing order of authoring contributions, for the 47 frontier journals, whose published articles were handled by at least 25 distinct editors. Women were underrepresented consistently across all fields and particularly severely in math-intensive disciplines.

The following figure supplements are available for figure 1:

**Figure supplement 1.** Analysis of network topology.

**Figure supplement 2.** Gender disparities vary between countries.

entire studied time-span (*Figure 2—figure supplement 1*).

The differences in assignment numbers may reflect behavioral or psychological differences between the groups of male and female scientists — either intrinsic or due to sociocultural context (*Moss-Racusin et al., 2012*; *Nature Neuroscience, 2006*; *Ceci et al., 2009*; *Ceci and Williams, 2010*; *Goulden et al., 2011*; *Ceci and Williams, 2011*; *Bloch, 2012*; *Raymond, 2013*; *Shen, 2013*; *Handley et al., 2015a*). Nevertheless, assignment numbers are also ultimately influenced by the editors' active choices. To reveal whether any bias exists in the reviewer assignment relation, we first analyzed gender correlations between directly connected pairs of nodes in the editor-to-reviewer appointment network (*Figure 3a*) and found a marked gender homophily bias for both male and female editor nodes. Specifically, 73% of reviewers appointed by men were also men, 33% of reviewers appointed by women were women, but, importantly, both these numbers laid above the expectations drawn from the assumption that genders were randomly distributed in the given editor-to-reviewer network topology. Similarly, in the reviewer-to-author network (*Figure 3b*), male (female) reviewers assessed articles authored by male (female) authors significantly more often than expected.

While these findings seem to point at homophily created by choices, they might also stem from "baseline" homophily (*McPherson et al., 2001*), i.e. subtle but unavoidable bias caused by disproportions in the number of reachable male and female nodes due to heterogeneous network structure. We first checked for the influence of local subnetwork structure on apparent gender bias by looking at different scientific fields, including those with relatively mild underrepresentation of women, and found homophily widespread across disciplines (*Figure 3c*). Second, a more detailed analysis of inter-node gender correlations in the editor-to-reviewer appointment network detected a clear tendency to gender homophily already at the level of the narrow neighborhood of individual nodes (*Figure 3d*). Specifically, to control for baseline homophily at the level of a narrow local neighborhood, we measured, for each editor node, the actual number of reviewer assignments given to women. We then subtracted from this number its chance expectation, derived individually for every node from the frequency of *locally reachable* female reviewers, i.e. reviewers situated at most five links away (which is a short distance relative to the average shortest path length of 12 steps for the editor-to-reviewer network, cf *Figure 1—figure supplement 1e*). Even at this local neighborhood level, we continued to find that male (female) editors generally appointed female reviewers at a lower (higher) rate than expected. Both independent analyses – by topic or localized – validate the existence of a so-called "inbreeding" homophily, i.e. an active preference to connect with same-gender network nodes, on top of "baseline" homophily (*McPherson et al., 2001*).

Finally, we wondered whether the observed inbreeding homophily in the network was due to the presence of a few strongly homophilic editors or whether, alternatively, homophilic attachment was a feature shared by most editors. To that end, we defined an index of inbreeding homophily at the local level of each editor node. For each considered editor node, we first evaluated the number $k$ of connected same-gender reviewers. We then evaluated the

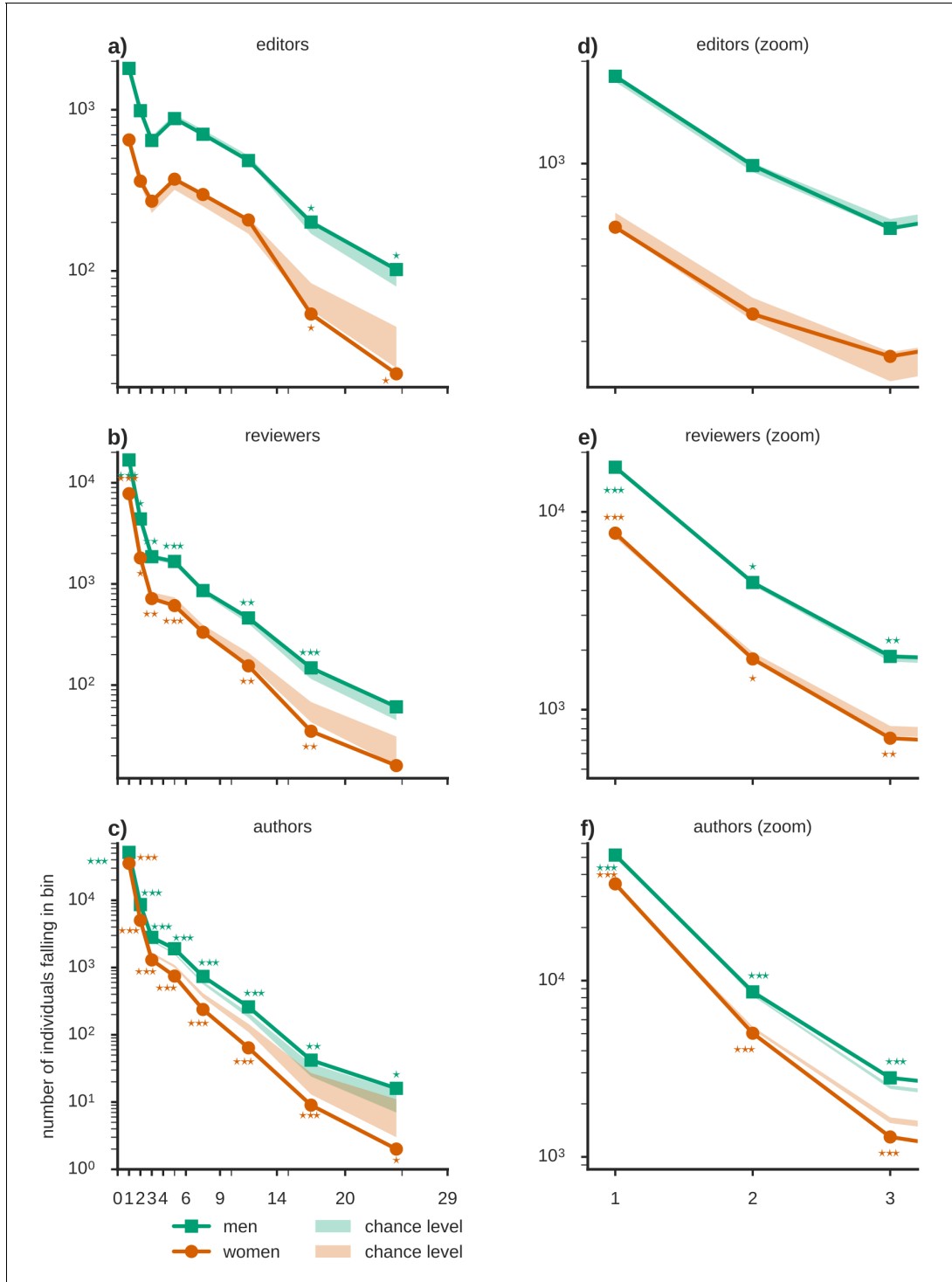

**Figure 2.** Women are underrepresented in the fat tail of contributions. A break-down of the number of individuals contributing a given number of times as editors, reviewers and authors (binned, x-axis is marking the bin edges) shows that the majority of scientists (**a**) edited, (**b**) reviewed or (**c**) authored (corresponding zooms for small contribution numbers are shown in **e-f**) only a small number of manuscripts. Chance levels (shaded) were derived from an ensemble of reference networks constructed as shown in *Figure 1a*. The underrepresentation of women in relation to these chance levels tends to increase towards the fat tail of the distribution, associated to the relatively few individuals that made many contributions. In the group of one-time authors or reviewers however, women are overrepresented. Time resolved distributions are shown in *Figure 2—figure supplement 1*.

The following figure supplement is available for figure 2:

*Figure 2 continued on next page*

*Figure 2 continued*

**Figure supplement 1.** Time- and gender-resolved histograms of the number of contributions.

probability $0 \leq \Phi_{hom} \leq 1$ that $k$ (or more) homophilic connections could arise by baseline homophily only, taking into account the editor-specific basin of locally reachable male and female reviewers (defined as for *Figure 3d*). Such $\Phi_{hom}$ can serve as an index tracking the strength of inbreeding homophily in shaping the actual reviewer appointments by an editor. Large values of $\Phi_{hom}$ approaching 1 indicate that the observed gender homophilic choices of a given editor are plausibly just due to "passive" baseline homophily. In contrast, small values of $\Phi_{hom}$ approaching 0 hint at a stronger tendency to "active" – consciously or unconsciously, see Discussion – inbreeding homophily. *Figure 3e* shows the histograms of the index $\Phi_{hom}$ for male and female editors, compared with expectations from gender-shuffled networks. For male editors, most histogram bins for $\Phi_{hom} < 0.6$ displayed node counts significantly larger than gender-shuffled estimations. The histogram of $\Phi_{hom}$ for female editors showed much fewer significant overrepresentation and most of them at very low values of $\Phi_{hom}$, however it remained compatible with gender-shuffled estimations for most of the $\Phi_{hom}$ range.

These different distributions of inbreeding homophilic tendencies resulted in a gender-dependent impact of the reviewer-appointment choices of male and female editors in determining the overall number of female reviewer appointments. To determine this impact we pruned links originating from editors with inbreeding homophily index $\Phi_{hom}$ below a growing threshold $\Phi_{thr}$ (retaining only editors whose $\Phi_{hom}$ satisfies $0 \leq \Phi_{thr} \leq \Phi_{hom} \leq 1$) and we did so separately for male and female editors (*Figure 3f*). After pruning the most homophilic male or female editors, we evaluated the new resulting probabilities of appointing a female reviewer. On the one hand, we found that it was enough to remove the few most homophilic female editors with the lowest values of $\Phi_{hom}$ from the network, to bring the probability for a female editor to appoint a female reviewer back to chance-level. On the other hand, the probability for a male editor to appoint a female reviewer increased only very slowly by pruning more and more male editors. In particular, it remained significantly below chance expectations for all the considered thresholds for inclusion, $0 \leq \Phi_{thr} \leq$ 0.5. This means that the overall smaller-than-chance probability of appointing female reviewers for male editors is due to inbreeding homophilic tendencies that are *widespread among male editors*, although at varying degrees of strength. In contrast, the overall larger-than-chance probability to appoint female reviewers for female editors is driven by the action of just a *small number of strongly homophilic female editors*, with most other female editors showing only "passive" baseline homophily.

## Discussion

In this study, we found that apart from a few outliers depending on country and discipline, women are underrepresented in the scientific community with a very slow trend towards balance, which is consistent with earlier studies (*Larivière et al., 2013*; *Fox et al., 2016*; *Topaz and Sen, 2016*; *Lerback and Hanson, 2017*; *Nature Neuroscience, 2006*; *Shen, 2013*; *Nature, 2012*). In addition, we found that women contribute to the system-relevant peer-reviewing chain even less than expected by their numerical underrepresentation, revealing novel and subtler forms of bias than numeric disproportion alone. We reported clear evidence for homophily beyond the expected baseline levels in both genders (*Figure 3*) using a very large trans-disciplinary data set that allowed us to clarify a previously ambiguous picture (*Lloyd, 1990*; *Gilbert et al., 1994*; *Borsuk et al., 2009*; *Buckley et al., 2014*; *Fox et al., 2016*). This network-level inbreeding homophily is driven by a large fraction of male editors, together with only a few highly homophilic female editors.

### Evolution of participation rates by gender and causes for remaining inequity

To start our discussion on a positive note, we found that the participation of women in science, at least in terms of their numerical representation, has increased during the last years, which is consistent with other studies. The number of female doctoral recipients at US institutions increased by, on average, 0.1% - 0.6% per year between 2005 and 2015, depending on broad field of study (*National Science Foundation, 2016*). *Ley and Hamilton (2008)* reported that the number of fraction of women in medical schools

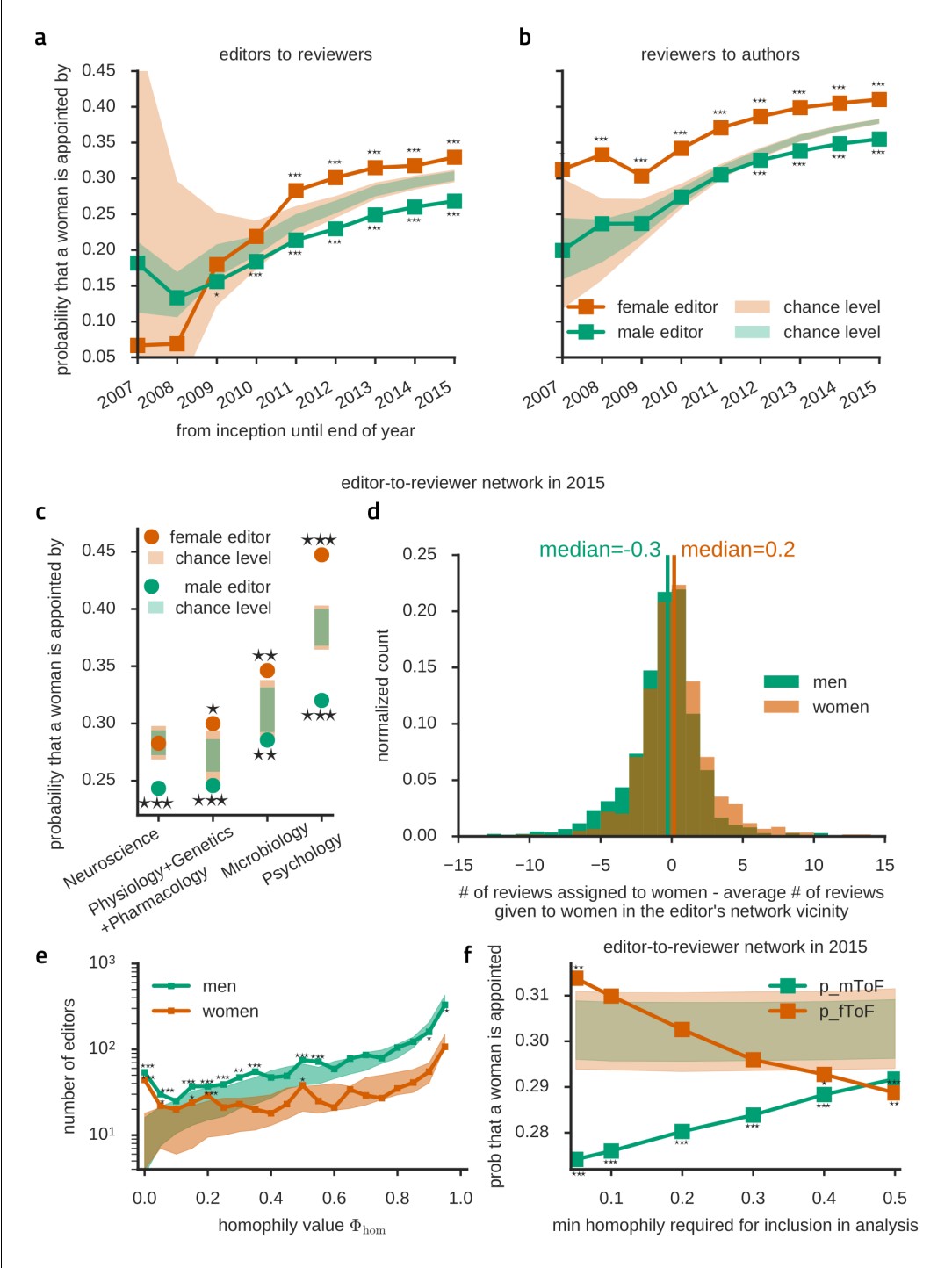

**Figure 3.** Editors have a same-gender preference for appointing reviewers. (**a**) Female editors (orange) appoint significantly more female reviewers than expected under the gender-blind assumption (shaded area). At the same time, male editors (green) appoint less women than expected. The development of this trend over time is shown, including articles cumulatively until the indicated year. (**b**) Likewise, female/male reviewers review significantly more female/male-authored articles than expected. (**c**) Homophily is widespread across scientific fields, including those with relatively mild underrepresentation of women. We here report four example disciplinary groupings, with large numbers of contributions (from left to right, respectively, 13416, 4721, 4020, 5680) and the propensity of appointing a female reviewer depending on the editor's gender for each of these groupings. Only assignments by female neuroscience editors were not homophilic, otherwise the occurrence of same-gender preferences was general, arguing against heterogeneity between subfields as a cause for homophily in assignments. (**d**) Plotted here are distributions of a measure of inbreeding homophily. To control for baseline homophily at the level of a narrow local neighborhood, we measure, for each editor node, the actual number of

*Figure 3 continued on next page*

Figure 3 continued

reviewer assignments given to women and subtract the expected number, which would be observed if the considered editor appointed women with the same frequency as in his/her local vicinity. For male editors (green) the distribution is skewed towards an underrepresentation of female assignments (left-leaning), while for female editors the distribution is skewed towards an overrepresentation of female assignments. This highlights that homophily bias is detectable even at the level of the reachable narrow surrounding of each editor. (e) Histogram of the probability that an editor assigns as least as many reviews to people of the same gender as he/she actually does reveals that there's an excess of strongly inbreeding-homophilic editors (small $\Phi_{hom}$ -values) among both men and women compared to expectation (shaded area). Note that below $\Phi_{hom} < 0.1$ there are only few strongly homophilic female editors. For male editors, significant homophily extends through many more editors until $\Phi_{hom} < 0.6$. (f) Using all data until 2015, the probability that a women is appointed is above expectation (shaded areas) only for female editors and only when all or all but the most extremely inbreeding-homophilic editors are included in the analysis.

increased by, on average, 0.6% to 0.8% per year (depending on the scientists' rank) between 1996 and 2007. Percentages of women professors has increased at a rate of 0.5%-1% per year in the European Union (*ETAN Expert Working Group on Women and Science,ETAN Expert Working Group on Women and Science, 2000*). The fraction of publishing female scientists in Germany increased by, on average 0.7% from 2010 to 2014, it is now 30.9% (*Pan and Kalinaki, 2015*). *Fox et al. (2016)* found that the number of selected female reviewers in Functional Ecology increased by, on average, 0.8% per year between 2004 and 2014, while, notably, the number of female editors increased by, on average, 3.8% per year. *Caplar et al. (2016)* noted that the number of female first authors of astronomy articles increased by about 0.4% per year between 1960 and 2015. In the Frontiers series of journals, we found that the number of contributions by female authors, reviewers and editors increased by, on average, 1.1% / year, 1.2% / year and 0.9% / year between 2012 and 2015, respectively, similar to the numbers above.

What could be the reasons for the remaining inequity? It has been argued that underrepresentation of women in science may be due to conscious career choices by female researchers (*Ceci et al., 2009*; *Ceci and Williams, 2010*, *2011*), even if it is not clear to which extent these choices are really free or rather constrained by society. Previous studies reported that, measured by their number of publications, women are generally less productive than men (*Cole and Zuckerman, 1984*; *Zuckerman, 1991*; *Long and Fox, 1995*; *Xie and Shauman, 1998*; *Pan and Kalinaki, 2015*; *Caplar et al., 2016*) and it has been suggested (*Xie and Shauman, 1998*) that this might be due to personal characteristics, structural positions, and marital status. Moreover, the fraction of female scientists decreases with rank or age (*ETAN Expert Working Group on Women and Science, 2000*; *Ley and Hamilton,*

*2008*; *Goulden et al., 2011*) and this shorter career length might contribute to the drop of female-to-male ratio for a high number of contributions. Nevertheless, women who persevere longer in their career despite obstacles are highly performing. While the productivity of young publishing female scientists in Germany was 10% lower than that of their male counterparts, the discrepancy reduced to just 3% for more senior scientists (*Pan and Kalinaki, 2015*). Also, it has been reported that women with children are not less productive than those without (*Hamovitch and Morgenstern, 1977*; *Cole, 1979*; *Cole and Zuckerman, 1987*), although young children might decrease productivity (*Kyvik, 1990*; *Kyvik and Teigen, 1996*). The low number of women among senior scientists might be particularly detrimental for a gender-neutral evaluation of scientific work, as the implicit association of "male" and "science" is strongest in the group of 40-65 year olds (*Nosek et al., 2007*). Moreover, declining an invitation to review is often due to a lack of time (*Tite and Schroter, 2007*) and it is possible that female scientists spend more time with duties beyond research (e.g. teaching, mentoring, service; *ETAN Expert Working Group on Women and Science, 2000*; *Knapp, 2005*; *Misra et al., 2011*). On the other hand, a compensating factor seems to be that female editors, in contrast to authors, have been reported to be more productive than male editors (*Gilbert et al., 1994*). Interestingly, men and women who are invited to review a manuscript have very similar propensities to accept the invitation (*Fox et al., 2016*; *Lerback and Hanson, 2017*), suggesting: (1) that simply increasing the number of invitations to female reviewers would have a direct and proportional effect; and, (2) that the low number of female reviews in our data is caused in part by a lower number of invitations.

The underrepresentation and discrimination of women in the scientific community is a problem

that will not solve by itself, given the pervasive, generally unconscious nature of gender bias. Women have been reported to be less likely to be hired (*Moss-Racusin et al., 2012*), to receive a grant (*Wennerås and Wold, 1997*), and to receive higher salaries (*Shen, 2013*). Still today, most people implicitly associate science with men, and liberal arts with women more than the other way round (*Nosek et al., 2007*), and this tendency, for both men and women, is apparent from a very young age (*del Río and Strasser, 2013*; *Bian et al., 2017*) and possibly reinforced by social dynamics in school education (*American Psychological association, 2007*; *Duru-Bellat, 2008*). Beyond that, men are more reluctant than women to believe that such a bias exists (*Handley et al., 2015a*), manifesting lack of interest for the problem ("negligence") or, even, consciously assuming that gender discrimination cannot be avoided ("philosophical acceptance") more often than females (*Parodi, 2011*).

### How representative are the Frontiers journals?

The data analyzed here comprises a wide spectrum of scientific topics and the findings should generalize. However, Frontiers articles are unusual insofar as they undergo open peer review, whereas the identity of reviewers is not revealed in most other journals. Ambiguous reports exist whether open-peer review (as opposed to single- or double-blind peer-review) affects potential reviewers' willingness to assess a paper (*Nature Neuroscience, 1999*, *van Rooyen et al., 1999*; *Ware, 2008*; *Baggs et al., 2008*). In particular, a primary concern in disclosing reviewer's identity is the possibility that a rejected author may also become a prospective employer for the reviewer and hence a possible reluctance of peers in more vulnerable positions to accept an invitation to review. While it is conceivable that assignment rejection due to non-anonymity is more likely for early career scientists, we do not see any reason for a direct effect of gender and such an effect has not been reported to the best of our knowledge.

Then, how does the population of scientists contributing to the Frontiers series of journals compare to other scientometric populations? First, we compared our authorship data to that of *Larivière et al. (2013)* who analyzed gender bias in articles from a wide range of journals that were published between 2008 and 2012, comprising about 3 million authors. While no analysis of peer review is performed therein, this study comprises an order of magnitude more authors than in our study. It can therefore serve as a benchmark for gender-composition among authors. They reported that 42% of authors in their analyzed scientific articles were women, whereas we found that number to be 39% in the Frontiers journals. Given uncertainties in determining a person's gender these numbers are comparable. Broken down by country, we find overall similar fractions of female authors, although, for some countries, the relative deviations can rise up to 29% (*Supplementary file 1*). However, small sample sizes, together with, possibly, a varying popularity of the Frontiers journals in different countries, might contribute to such deviations.

Second, not much data was available concerning gender bias among reviewers and editors, until very recently. Many previous studies (cf. *Supplementary file 2*) were self-diagnoses performed by editorial boards of the corresponding journal and, as a consequence, tended to be based on mono-disciplinary data of relatively small sample size. Larger sample sizes, but limited to editors, were considered in an analysis of the composition of editorial boards of 435 mathematical journals (*Topaz and Sen, 2016*). Only 9% of editors were women. Other reported numbers for the fraction of women editors in journals of different disciplines range from 38% to 54% (cf. *Supplementary file 2*). These numbers lie at the lower and upper end of the female editor fractions across the Frontiers journals, ranging between 6% (Frontiers in Robotics and AI) and 37% (Frontiers in Aging Neuroscience), with an average of 28%. Concerning female reviewers or female reviewer appointments, fractions reported in the literature range between 16% and 48% (cf. *Supplementary file 2*), to be compared with the range between 11% and 48% for the Frontiers journals with an average of 30%. Concerning female authors, *Pan and Kalinaki (2014)* report fractions ranging from 15% in computer science to 57% in veterinary science. These numbers are once again comparable to female author fractions in Frontiers journals, ranging from 17% (Neurorobotics) to 48% (Public Health). Overall, our study provides thus a global account on the prevalence of women among editors and reviewers and ranks previous reports in a continuum of field-specific participation numbers. Importantly, our data is consistent with these diverse reports, highlighting that the Frontiers peer-review networks are well representative of widespread patterns.

Our work calls for a detailed comparison with another recently published report about peer reviewer assignments in 20 journals of the American Geophysical Union (AGU), based on a slightly smaller sample size compared to ours (*Lerback and Hanson, 2017*). This study reports information about aspects that our study could not have access to, breaking down women's underrepresentation by age and showing that the decline rate for invited reviews is only slightly smaller for women than men. Overall, relative fractions of female participation reported by this study are compatible with numbers we found for the journal Frontiers in Earth Science, with e.g. a matching female reviewer appointments fraction close to 20%, suggesting that women play a larger role in other fields compared to that report (cf. *Figure 1c*). For the AGU journals the authors conclude that editors, especially male ones, appoint too few female reviewers. Male editors' behavior in that study thus agrees with our findings for the entirety of Frontiers journals, while we find an opposite trend for female editors. We note here that *Lerback and Hanson (2017)* reached their conclusion of women's underrepresentation by comparing actual reviewer appointment numbers to the fraction of female first authors. This comparison, however, might be questionable, because reviewers in low age groups are rarely invited by editors (3% of times) whereas first authors tend to be young. To account for such differences, we determined expectation levels by gender shuffling among the reviewers and editors in the fixed network of actual reviewer-editor interactions and find that the fraction of female authors (the expectation value that Lerback and Hanson used) is much higher than the expected number of female reviewer contributions (our expectation value; cf. *Figures 1b* and *3a*). For that reason, Lerback and Hanson may have quantitatively overestimated the female editors' bias against female reviewer appointments. Still, despite this overestimation, even Lerback and Hanson reported female editors' preference for female reviewers for certain age classes (although not commented upon).

## Homophily in society and science

The phenomenon of gender homophily in peer-reviewing networks have already been described, but these previous reports have reached ambiguous conclusions. *Lloyd (1990)* found that female reviewers accepted female-authored papers at a higher rate than those of male authors, whereas male reviewers did not show such a bias. In contrast, *Borsuk et al. (2009)* reported that male and female reviewers were equally likely to reject a female-authored paper. The probability that a female editor appoints a female reviewer was reported to be 31%-33%, whereas male editors appointed female reviewers in 22%-27% of cases (*Gilbert et al., 1994*; *Buckley et al., 2014*; *Fox et al., 2016*). Here, for the whole spectrum of Frontiers journals we found these numbers to be similar: 33% and 27%, respectively. However, our study concludes for the existence of significant inbreeding homophily in the reviewer appointing behavior for both male and female editors, and does so based on a pluri- rather than mono-disciplinary data set, substantially larger than all previous accounts of homophily in peer review.

Socrates, in Plato's Phaedrus, already asserted that: "similarity begets friendship". Homophily – or "attraction for the similar", not only limited to the gender attribute – is ubiquitous in social networks. Since the classic studies of *Park and Burgess (1921)* and *Lazarsfeld and Merton (1954)*, gender homophily has been found in groups of playing children (*Bott, 1928*; *Shrum et al., 1988*) and adult friends (*Verbrugge, 1977*) and is also present in work environments (*Brass, 1985*; *Bielby and Baron, 1986*; *Ibarra, 1992*) and voluntary organizations (*Popielarz, 1999*). Since focused interactions between co-workers favor the formation of relations, operation in already homophilic environments will lead to an amplification of homophily (*Feld, 1981*; *Feld, 1984*). In particular, homophilic styles of professional interaction with peers may persist since the time in which they were (un-)consciously learned in homophilic school environments (*Vinsonneau, 1999*).

Importantly, even a slight homophily can influence and alter the way in which information spreads (*Yava and Yucel, 2014*) and opinions form through the social network of interactions, leading to the emergence of "dead-end" cultural niches (*Mark, 2003*). Homophilic groups indeed tend to vote together when asked to decide for something (*Caldeira and Patterson, 1987*) and have similar prospective evaluations, a same mindset (*Galaskiewicz, 1985*). While homophily can in principle be put to good use, as for instance in the education about good health practices (*Centola, 2011*), the uncontrolled effects of homophily may constitute a

threat to the universalism of the peer-review system, and thus to science.

### Gender-specific mechanisms of homophily

We observed very different patterns of homophily for male and female editors, with a widespread homophily across men, while dominated by very few highly homophilic editors for women. After removal of their contribution, homophily became insignificant (cf. *Figure 3e,f*). This suggests that there is only baseline homophily for the majority of female editors and most assignments are gender-blind (for instance in the neuroscience community, cf. *Figure 3c*). Differences between men's and women's homophily patterns are classically known, finding their root in different styles of social network construction. For instance, in situations where a mutual friendship exists between A and B a friendship initiation with C tends to be reciprocated by boys, but not by girls (*Eder and Hallinan, 1978*). Such differences in attachment strategies tend to generate gender-segregated worlds for children to preadolescents in which girls evolve in small homogeneous groups and boys form larger but more heterogeneous cliques, with boundaries made looser only later by romantic ties (*Shrum et al., 1988*). Professional social networks of men are more homophilic than women's, especially in work environments in which men are dominant (*Brass, 1985*; *Ibarra, 1992*). Another source of asymmetry may be that both men and women tend to form connection routes passing through a male node when reaching toward distant domains (*Aldrich, 1989*).

One could speculate that other factors might contribute, like friendship or (perceived) status, competency and reputation. These factors might, in turn, be partly depending on gender, e.g. through implicit biases (*Nosek et al., 2007*; *Merton, 1968*; *Paludi and Bauer, 1983*). Multiple categories of relationships were analyzed, for instance, by *Ibarra (1992)* who reported that, in a company setting, men named mostly men as points of contact for five different business-relationship categories, whereas for women the preferred gender was category-dependent. A similar situation could be at work here: one could speculate that a set of other, hidden, variables influence reviewer appointment decisions, and that these variables have a different importance for male and female editors. Determining which factors are most important for male and female editors in the choice of the reviewer and how these factors are or are not, in principle, related to gender, might thus aid in reducing homophily in the peer-review system.

Our finding of strongly homophilic "topology-organizer" female editors is reminiscent of the notion of "femocrat" introduced in political studies, referring to the role played by isolated feminists who, after having managed to integrate inside men-dominated decisional organisms, provide a bridge to the spheres of power for the requests of activists outside of them (*Yeatman, 1990*). Now, while the active engagement of these femocrats is very useful in pushing forward technocratic (i.e. top-down) solutions aiming at reducing gender discriminations, especially at an early stage, on the long-term, the effects of their action may be precarious. Indeed political experiences have shown that when an external event reduces the influence of these isolated driver women, the situation can quickly deteriorate again (*Outshoorn, 2005*), aggravated by the suspicious look toward femocrats held by formerly dominant men or, paradoxically, even women, finding them too prone to compromise or too aggressive (*Outshoorn and Kantola, 2007*). It is thus important to devise strategies 'healing' network topology in depth, and in a bottom-up fashion, via pervasive education campaigns targeted to the deciders (*Sainsbury, 1994*), in our case chiefly the editors. Such strategies are required to protect the acquirements of top-down actions against gender discriminations: increasing the number of women will not be enough to overcome gender bias (*Isbell et al., 2012*; *Avin et al., 2015*).

## Conclusions

Ideally, all scientific interactions are gender-blind. A scientist's status and the provision of resources to scientists should not be influenced by gender but solely depend on the value of the scientific contributions. Access to the publication systems is a critical determinant of a scientist's success. Accordingly, reviewers and editors, the gatekeepers of the scientific canon, should be particularly sensitive to base their judgment solely on the merit of scientific work. This merit, however, is difficult to determine and any assessment is necessarily influenced by the assessor's view of the field, including his or her personal position in the network of colleagues and the interactions with them (*Mulkay, 1979*; *Cole, 1992*).

Inbreeding homophily, an increased affinity between persons with similar attributes, appears to be a sociological, population-level trait of

human societies. It is only natural, thus, that we find gender homophily in interactions between editors, reviewers and authors. Nonetheless, this inbreeding homophily is damaging to female scientists, whose work ends up being overlooked, due to unconscious negative bias. The phenomenon of inbreeding homophily is also likely not restricted to the peer review of manuscripts, so it needs to be taken into account for grant evaluation, hiring, or when designing mentoring programs. Importantly, it is likely to persist even when numerical balance between genders is achieved (*Isbell et al., 2012*). Altogether, inbreeding homophily negatively affects science as a whole because a stronger involvement of women would increase the quality of scientific output (*Merton, 1973*; *Woolley et al., 2010*; *Nature, 2013*; *Campbell et al., 2013*). Consequently, all scientists should wholeheartedly support the endeavor to remove gender bias from science - but how could that be achieved?

Initiatives to remove gender-based inequality can roughly be divided into two different categories. On the one hand, "gender mainstreaming" (*Special Adviser on Gender Issues and Advancement of Women, 2002*) promotes the consideration among actors at all levels of every action's and policy's implications on women and men and is geared towards creating long lasting "bottom-up" changes. On the other hand, fast progress could be attempted through "top-down" implementation of technocratic instruments such as quota. This politically issued 'state feminism' (*Mazur and McBride Stetson, 1995*), is suboptimal in that it might even "provide an alibi" for not modifying attitudes in depth (*Squires, 2008*). As inbreeding homophily is an expression of a state of mind it is likely little amenable to change by externally enforced measures. Raising awareness, in comparison, seems to be the most promising route. The goal should be to motivate all scientific actors to "integrate thinking about gender discrimination in every decisional process" (translated from *Woodward, 2008*). Educative actions should be conducted with tact, not based uniquely on inducing feeling of guilt and shame, in order not to be perceived as annoying (*Woodward, 2003*). At the same time, existing formal actions to reduce bias should be upheld.

In the field of peer review two more specific strategies are available to reduce bias: blind review and automated editorial management. However, both strategies are of limited acceptance and use. First of all, removing the authors' names is often not sufficiently blinding.

References to the authors' previous publications or to the approving ethics committee all but spell out the authors. Second, while removal of the authors' names does indeed blind the reviewers to all irrelevant attributes, it also blinds them to relevant meta-data, such as the scientific experience of the authors, which might be considered as relevant by many reviewers. In an attempt to assist editors of Frontiers journals, keyword-based reviewer suggestions are automatically provided to them but the editors remain free to make their own choices. While these gender-blind automated suggestions could already contribute to an assignment that is less influenced by homophily, an editorial management software is also the ideal platform to routinely direct the editor's attention to the issue of homophily. It could display statistics similar to our *Figure 3* and encourage non-homophilic choices of reviewers. Such a strategy maintains full editorial freedom and could easily be evaluated, either internally or, in the case of open review as in the Frontiers journals, through analysis of the publicly available data.

Given how engrained homophily is in our nature, the path towards a gender-blind science will be arduous. Yet, with the joined effort of the scientific community to overcome partisanship and discrimination, a merit-based system with equal opportunities for all scientists might just be within reach. After all, which social enterprise would be more apt to follow ratio over instinct than science?

## Materials and methods

### Collection and parsing of data

All article data were exclusively obtained from the publicly available articles web pages from the Frontiers Journal Series (RRID:SCR_007214), which was listed (at the date of last data download in March 2016) on: http://www.frontiersin.org/SearchIndexFiles/Index_Articles.aspx, as well as the associated XML file if the HTML code of the article web page contained a corresponding reference. Subsequently, articles' metadata (article id, authors, reviewers, editors, publication date, etc.) were extracted from the XML files and the web pages. All gathered personal identity information was deleted after inference of individual genders (see later), resulting thus in a fully anonymized data set. In total, we analyzed 41'100 articles published before January 1st, 2016, covering 142 Frontiers journals from Science, Health, Engineering, and Humanities and

Social Sciences. Our parsing routine was able to find information about authors in 41'092 of these articles, about reviewers in 39'788 articles (note that some articles, like editorial articles, might not have been reviewed), and about editors in 40'405 articles. The anonymized network data is provided as *Supplementary file 3*.

To recognize and identify people re-occurring in more than one article, every person was assigned a unique identifier number (UID). When a contributor was found to be associated to an official profile identification number in the Frontiers database, then we relied on it, directly translating it into a UID (this happened for 71% of contributors). In the remaining cases, we decided whether a record matched another based on the names and affiliations of people. Specifically, for two names to be matched, we required that the surnames coincided and that each given name of the contributor with less given names needed to have a corresponding match in the other contributor's name (a match could also be an initial like "J" with a fully specified name like "John"). In case both contributors' given names consisted of only initials, we required, in addition, that their affiliations were sufficiently similar. *Newman (2001)* found that name-matching in the absence of UIDs, and even abbreviating all given names to initials, resulted in errors on the order of few percent in a data set comprising more than a million people. Correspondingly, as we expect the UIDs to be correctly associated with a contributor in the vast majority of cases, erroneously matching or not matching people is likely relatively uncommon.

### Determination of gender

Each UID was assigned a gender based on their associated given names (note that after the steps described in the previous section, at least one first name was fully specified for 99.6% of the UIDs, while for the remaining 0.4% of UIDs all given names consisted of only initials so that no gender could be attributed). The extracted given names were compared with an extensive name list, assembled from public web-sources, such as:

- http://japanese.about.com/library/blgirls-name_[a-z].htm,
- http://japanese.about.com/library/blboys-name_[a-z].htm (retrieved December 9, 2015)
- http://www.top-100-baby-names-search.com/chinese-girl-names.html,

- http://www.top-100-baby-names-search.com/chinese-boys-names.html
- http://www.babynames.org.uk (retrieved December 11, 2015)
- US census data (https://www.ssa.gov/oact/babynames/limits.html; retrieved March 17, 2016).

Note that some given names (like Andrea) are in use for both men and women. Gender-ambiguous given names present in the US census database were categorized to the gender to which they were more frequently attributed. When a name appeared as both male and female in one of the other sources, or when different sources did not agree on the gender for a name, we decided not to associate that given name with a gender.

We validated the gender assignment procedure by performing a web search for 1053 randomly selected people from our data set, and determining their gender based on a picture or the use of gender-specific pronouns in a biographical text. We were able to find such information for 924 out of the 1053 people (88%). The gender automatically assigned by our algorithm to those identified was correct in 96 % of cases. For comparison, we note that the name-gender algorithm used in *Larivière et al. (2013)* misclassified male and female names in 8% of cases.

Our list thus comprised 66605 female and 43482 male names. In addition to the name list, we manually assigned the non-automatically-identified gender of 643 people with a high number of re-occurrences. In total, we were thereby able to assign gender to 131885, that is 87 % of UIDs. All further analyses were done ignoring the remaining 13% of scientists.

### Network construction

We represented the available data in directed networks (*Figure 1a*), in which vertices were individual scientists and edges denoted peer-reviewing interactions: is appointing in the editor-to-reviewer network, and is editing (reviewing) a manuscript of in the editor (reviewer)-to-author network. Year-resolved graphs were constructed by deleting all links representing articles that were published later than the given year.

### Graph analytics

All graph analyses (*Figure 1—figure supplement 1*) were performed with the freely-available Python igraph package.

In graph theory, a connected component is a subgraph in which any two vertices are connected to each other by at least one path, and which is connected to no additional vertices in the full graph. The largest of all the connected components of a graph is called its giant component. One can distinguish between the weak giant component (in which the direction of edges is ignored when building inter-node paths) and the strong giant component (in which the direction is taken into account). All the following graph analyses have been performed on the weak giant component of the networks observed at each time.

Transitivity undirected (clustering coefficient) is calculated as the ratio of triangles to connected triangles (triplets) in the graph, considering connections between nodes independent of their direction.

Average path length calculates the mean of the geodesic directed path lengths between all pairs of nodes in a connected component. The geodesic path length between a given pair of nodes is the minimum number of links needed to travel between the nodes along connected edges.

Small-worldness $S$ is defined in *Humphries and Gurney (2008)*, as $S=\gamma/\lambda$. $\gamma$ is the undirected transitivity of the graph divided by $k/n$, which is an approximation for the undirected transitivity of an Erdös-Rényi random graph with $n$ nodes and average degree (in+out) of $k$. $\lambda$ is the ratio of the average shortest path length of the graph to $\ln(n)/\ln(k)$, which is the average shortest path length of an Erdös-Rényi graph with $n$ nodes and average degree $k$.

### Statistical testing

Statistical significance was established by comparing a feature of the data to its confidence interval (CI). The graphic notations *, ** and *** denote that this feature lay outside the 95%, 99% and 99.9% CI, respectively. Confidence intervals were calculated by recalculating the given feature 10000 times, after permuting gender labels (with the exception of *Figure 3*e where, for computational reasons, only 100 recalculations were performed). Specifically, *Figures 1* and *2* are derived from a table with a column given the number of contributions (up to a specified time point) in a given role for each person, and another column of each person's gender, and the latter column was permuted keeping the former constant. On the other hand, confidence intervals in *Figure 3* were obtained by repeatedly permuting genders among all nodes in a given graph, independent

of their associated roles. The underlying graph used for *Figure 3a* and *Figure 3c-f* was a suitably pruned editor-to-reviewer graph, out of which: we first removed all self-loops (i.e. editor and reviewer are identical); second, we deleted all leaf nodes, i.e. scientist who never edited or reviewed anything and had therefore a null out-degree; third, for *Figure 3c*, we removed cross-disciplinary assignments from journals not belonging to the indicated category. Similarly, *Figure 3b* was derived from a deleafed reviewer-to-author graph.

### Inbreeding homophily at a local level

*Figure 3d* shows two histograms, one over all male editor nodes, the other over all female editor nodes. For each editor $i$ who appointed at least 2 distinct reviewers we calculated a measure $H_i$ of inbreeding homophily. To compute it, we first measured the actual number of reviewer assignments given to women nodes by the considered editor $i$, $W_i$. The next step was to subtract the expected number of reviewer assignments given to women, which would be observed if the given editor node appointed women with the average frequency $p_i$ they are appointed in its local vicinity. To evaluate $p_i$ we took the set of all editors (both males and females) at a distance of at most 5 directed edges from the considered editor node $i$. We counted the overall number $A_{all}$ of reviewer assignments made by these editors (i.e. the total number of edges originating from editor nodes in the neighborhood shell), and neglected those editors for which $A_{all} < 62$ (i.e. we required that, on average, at each of the 5 steps away from the considered editor at least 2 novel reviewers are encountered that could not have been reached in a shorter step count). We then determined the number $A_{female} \leq A_{all}$ of reviewer assignments made toward female nodes. We finally assumed $p_i = A_{female} / A_{all}$.

We could then compute the local inbreeding homophily measure $H_i=W_i - A_i\, p_i$, where $A_i$ was the total number of assignment made by each considered editor $i$.

We used a similar technique to assess the impact of the most homophilic editors on the overall network-homophily in the female editor-to-reviewer and male editor-to-reviewer networks. Let $q_i$ be the probability a person of the same gender is chosen by an editor $i$, where $q_i$ is calculated exactly as $p_i$ in the previous paragraph, i.e. by considering all people at most 5 directed edges away from editor $i$ in the editor-to-reviewer network, counting the number of

assignments these people gave to people of the same gender and dividing by the total number of assignments these people made. Next, let $k_i$ denote the number of assignments editor $i$ gives to a person of the same gender and $n_i$ the total number of assignments editor $i$ makes. Assuming editor $i$ chooses the gender of a reviewer at random, the probability that $i$ assigns $k_i$ out of the $n_i$ reviewers to have the same gender follows a binomial distribution binom $(k_i; n_i, q_i)$ and $\Phi_{hom}$

$$= \sum_{\nu=k_i}^{n_i} \mathrm{binom}(\nu; n_i, q_i)$$ measures how likely it is that

editor $i$ assigns at least $k_i$ reviews to a person of the same gender.

**Markus Helmer** is in the Max Planck Institute for Dynamics and Self-Organization, Göttingen, Germany, the Bernstein Center for Computational Neuroscience, Göttingen, Germany, and Yale University, New Haven, United States

markus.helmer@yale.edu

http://orcid.org/0000-0001-9680-0595

**Manuel Schottdorf** is in the Max Planck Institute for Dynamics and Self-Organization, Göttingen, Germany, and the Bernstein Center for Computational Neuroscience, Göttingen, Germany

http://orcid.org/0000-0002-5468-4255

**Andreas Neef** is in the Max Planck Institute for Dynamics and Self-Organization, Göttingen, Germany, and the Bernstein Center for Computational Neuroscience, Göttingen, Germany

**Demian Battaglia** is in the Bernstein Center for Computational Neuroscience, Göttingen, Germany, and the Institute for Systems Neuroscience, Aix-Marseille University, Marseille, France

demian.battaglia@univ-amu.fr

http://orcid.org/0000-0003-2021-7920

*Author contributions:* MH, Conceptualization, Data curation, Software, Formal analysis, Validation, Investigation, Visualization, Methodology, Writing—original draft, Writing—review and editing; MS, AN, Conceptualization, Formal analysis, Validation, Investigation, Methodology, Writing—original draft, Writing—review and editing; DB, Conceptualization, Formal analysis, Validation, Investigation, Methodology, Writing—original draft, Project administration, Writing—review and editing

*Competing interests:* The authors declare that no competing interests exist.

## Additional files

### Supplementary files

• Supplementary file 1. Comparison of female author contributions by country between *Larivière et al. (2013)* and the Frontiers series of journals.

• Supplementary file 2. Reported fractions of female authors, reviewers and editors in previous studies.

• Supplementary file 3. Network data. Contains two files, one (graph_nodes.csv) giving a random ID per person in the network together with that person's gender, the other (graph_edges.csv) indicating which IDs interact together with the roles of each ID (a: author, r: reviewer, e: editor).

### Funding

| Funder | Grant reference number | Author |
|---|---|---|
| Bundesministerium für Bildung und Forschung | 01GQ1005B | Andreas Neef Demian Battaglia |
| Marie Curie Career Development Fellowship | FP7- IEF 330792 (DynViB) | Demian Battaglia |
| Boehringer Ingelheim Fonds | | Manuel Schottdorf |

The funders had no role in study design, data collection and interpretation, or the decision to submit the work for publication.

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
