## [Decision Letter]

Thank you for submitting your article "Gender bias in scholarly peer review" to *eLife* for consideration as a Feature Article. Your article has been reviewed by three peer reviewers, and the evaluation has been overseen by a Senior/Reviewing Editor. The reviewers have discussed the reviews with one another and the Reviewing Editor has drafted this decision to help you prepare a revised submission.

Summary:

This is an interesting and high quality manuscript on a topic that is important and timely. The manuscript (which is based on a novel data set) merits publication after revision to improve readability and replicability.

Essential revisions:

1) To maintain high standards of research reproducibility, and to promote the reuse of new findings, *eLife* requires all major data sets associated with an article to be made freely and widely available (unless there are strong reasons to restrict access, for example in the case of human subjects’ data), in the most useful formats, and according to the relevant reporting standards. (See below for more details).

2) How representative is the population studied by the authors? It would be interesting to compare the population studied by the authors with other large-scale studies to enhance the generalizeability of their findings. For example, it seems as if the authors could compare Figure 1—figure supplement 2 to Lariviere et al. (2013) in order to determine whether the population they study is significantly different to traditional scientometric populations. Also, there are several studies that show that women are less productive than men. This might be useful for explaining Figure 2.

3) The authors should more fully acknowledge the limitations of the data set. For example, the authors mention that the names are publicly disclosed (this is what allows this analysis). However, several studies have shown that open peer review significantly affects the willingness of certain populations to conduct reviews (e.g., by age, rank, gender, minority status). Could the fact that names are disclosed make this less generalizable? The limitations of this should be discussed.

4) The analyses only included reviewers who agreed to review (understandably, as there is no publicly available data for reviewers who declined to review). However, women might be more likely to say no to requests to reviews for various reasons (e.g. they might have less time to review, or they might feel less qualified to comment on their peers' work). Hence, might their underrepresentation stem from self-selection rather than discrimination?

5) It would be interesting to comment on the possible effect of age on gender discrepancies at different levels. Specifically, editors might (on average) be older than reviewers, who in turn might (on average) be older than authors (although with large variability). Is it possible that higher gender discrepancy among editors and reviewers is because women might be less represented in older age groups?

6) There is sparse information on author name disambiguation (for those that lacked a UID). More information should be provided on this. The authors state that they cannot guarantee a "completely error-free parsing" – could they run a validation test to give an estimate of the error? And was any validation exercise conducted on the name-gender algorithm? How does this compare with other name-gender algorithms?

7) Several seminal books on peer review and articles specifically focused on bias in peer review are omitted here. Furthermore, the article fails to engage with the previous literature in a substantial way. Given that this work is one of the largest peer-review studies to-date, it would be useful for the authors to establish how their work refutes or confirms previous work.

8) Regarding the third paragraph in the Results section: please include a figure which shows that the gender gap in various contributions cannot be fully explained by population.

9) There are many interesting results that warrant more explanation in the text. At present, some of the most interesting results are buried in the figure captions. It would be great to see the authors embed some of these results, in more detail, in the main text.

10) Overall, the differences between observed and chance-level participation of women (for instance in Figure 2) are not that large, even though they are statistically significant. How concerned should we be about them in the first place?

11) The authors advise that editors should be encouraged to make non-homophilic reviewer selection. But if the goal is to increase gender parity, shouldn't women editors be 'allowed' to be more homophilic, at least until the parity achieved? Might it be better to suggest that all editors should try to recruit more women reviewers?

---

## [Author Response]

*Essential revisions:*

*1) To maintain high standards of research reproducibility, and to promote the reuse of new findings, eLife requires all major data sets associated with an article to be made freely and widely available (unless there are strong reasons to restrict access, for example in the case of human subjects’ data), in the most useful formats, and according to the relevant reporting standards. (See below for more details).*

We thank the reviewer for the reminder. We have included the anonymized raw data in the revised submission.

*2) How representative is the population studied by the authors? It would be interesting to compare the population studied by the authors with other large-scale studies to enhance the generalizeability of their findings. For example, it seems as if the authors could compare Figure 1—figure supplement 2 to Lariviere et al. (2013) in order to determine whether the population they study is significantly different to traditional scientometric populations. Also, there are several studies that show that women are less productive than men. This might be useful for explaining Figure 2.*

We thank the reviewer for these suggestions. We have now augmented the Discussion section with a comparison of the Frontiers population with other scientometric populations (subsection “How representative are the Frontiers journals?”) and included two tables ([Supplementary-material SD1-data] and [Supplementary-material SD2-data]) summarizing previous studies. Specifically, we find that gender composition of Frontiers articles authors is similar to that of Lariviere et al. (2013), although when broken down by country outliers do exist. We also included a comparison with (1) a large data set for the gender composition of editors in mathematical journals (Topaz and Sen, 2016) and (2) a recent study (Lerback and Hansen, 2017) with data from 20 journals of the American Geophysical Union. We find that, overall, our study provides a global account on the prevalence of women among editors and reviewers and ranks these previous reports in a continuum of field-specific participation numbers.

While our data does not allow inferring the reasons for why women contribute less, some studies indeed suggested that women are less productive. These results are now mentioned and discussed as a potential reason in the revised Discussion (subsection “Evolution of participation rates by gender and causes for remaining inequity”, second paragraph). However, for completeness, we also refer to other partially conflicting studies indicating that women may have different, not necessarily lower productivity patterns than men (e.g. Gilbert et al., 1994; Pan & Kalinaki, 2015). Furthermore, we stress that the key new findings of our study concern homophily (Figure 3) and that the smaller number of women has been taken into account in the analysis leading to these findings.

*3) The authors should more fully acknowledge the limitations of the data set. For example, the authors mention that the names are publicly disclosed (this is what allows this analysis). However, several studies have shown that open peer review significantly affects the willingness of certain populations to conduct reviews (e.g., by age, rank, gender, minority status). Could the fact that names are disclosed make this less generalizable? The limitations of this should be discussed.*

We thank the reviewer for pointing this out. We agree that open peer review could potentially have effects other than the intended increase in transparency and quality. These effects are now explicitly discussed in the revised manuscript (subsection “How representative are the Frontiers journals?”, first paragraph). Despite extensive search we could not find any study that specifically assesses whether open peer review affects the gender specific acceptance rate of reviewer assignments. Should we have missed a crucial publication, we would be very grateful for a reference.

We also now systematically compare our findings to a large number of previous studies which considered smaller sample-sizes but had access to more meta-data, since they were initiated journal-side. Overall, as we comment in the extended Discussion, we find women’s participation rates among editors, reviewers and authors of Frontiers journals to be comparable to previous reports, indicating that the Frontiers community is well representative of a general situation.

*4) The analyses only included reviewers who agreed to review (understandably, as there is no publicly available data for reviewers who declined to review). However, women might be more likely to say no to requests to reviews for various reasons (e.g. they might have less time to review, or they might feel less qualified to comment on their peers' work). Hence, might their underrepresentation stem from self-selection rather than discrimination?*

The reviewer is correct in suggesting that it is critical to distinguish between self-selection and discrimination. Our data set does not allow inferring the reasons for women’s underrepresentation and therefore self-selection (due to various motivations) could potentially play a role. This possibility, along with others, is now acknowledged and discussed in the revised manuscript (subsection “Evolution of participation rates by gender and causes for remaining inequity”, second paragraph). Besides this discussion, we also performed a more detailed analysis of the different prevalence of homophily in the male and female subpopulation. Taken together we feel that the data clearly point to same-gender preference of mostly male editors as a contribution to the under-representation of female scientists across the Frontiers series of journals. This conclusion is particularly strengthened by our new analysis of the different patterns of gender homophily for male and female editors (Figure 3-f).

*5) It would be interesting to comment on the possible effect of age on gender discrepancies at different levels. Specifically, editors might (on average) be older than reviewers, who in turn might (on average) be older than authors (although with large variability). Is it possible that higher gender discrepancy among editors and reviewers is because women might be less represented in older age groups?*

Our data set does not include scientists’ ages and we are, therefore, unable to analyze how age affects women’s participation in the peer-review system. However, as the number of female scientists decreases with age and rank, the differences seen in the gender fraction of editors, reviewers and authors could well be influenced by the age distribution in these populations. The revised manuscript now mentions this possibility (subsection “Evolution of participation rates by gender and causes for remaining inequity”, second paragraph) and we further discuss our results in light of a recent study with access to age data (Lerback and Hansen, 2017; subsection “How representative are the Frontiers journals?”, last paragraph).

*6) There is sparse information on author name disambiguation (for those that lacked a UID). More information should be provided on this. The authors state that they cannot guarantee a "completely error-free parsing" – could they run a validation test to give an estimate of the error? And was any validation exercise conducted on the name-gender algorithm? How does this compare with other name-gender algorithms?*

We thank the reviewer for pointing this out. The gender assignment is a critical step of the analysis and we have now included both a more detailed description of the name-gender algorithm and a validation test in the Materials and methods section (subsections “Collection and parsing of data” and “Determination of gender”). While errors are in principle unavoidable when identifying people based on their names, Newman (2001) estimated their rate to be only few percent. Combined with our reliance on user IDs for the vast majority of people, we expect that errors due to name disambiguation are uncommon.

To estimate the error rate in gender assignment, we have validated the genders of a random selection of 924 scientists through a manual web search (as described in subsection “Determination of gender”, second paragraph) and compared our assignment error with the supplemental material of Lariviere et al. (2013). Notably, our algorithm’s error rate is smaller than the reported error rate of the name-gender algorithm of Lariviere et al. (2013). Independent of the error rate, we would like to stress that mis-assigning gender can never increase the homophily that we see. Random mis-assignments have the same effect as a small rate of random permutation of gender labels. This would reduce evidence for inbreeding homophily rather than spuriously inducing it Thus, although we cannot reach a 100% correct gender assignment, the finding of inbreeding homophily is robust against this type of uncertainty.

*7) Several seminal books on peer review and articles specifically focused on bias in peer review are omitted here. Furthermore, the article fails to engage with the previous literature in a substantial way. Given that this work is one of the largest peer-review studies to-date, it would be useful for the authors to establish how their work refutes or confirms previous work.*

We thank the reviewer for pointing this out. It is true that our previous version had the style of a short letter, rather than that of an extended article. We now rewrote and substantially extended the Discussion section of the manuscript, including generous paragraphs of comparison with other empirical work, the current state of the art and detailed discussions about why our study can clarify previously ambiguous pictures.

Despite these comprehensive additions and expanded reference section (99 references cited, while in the previous version we had just 31) we have failed to include the crucial references the reviewer had in mind; we would be keen to receive more explicit guidance.

*8) Regarding the third paragraph in the Results section: please include a figure which shows that the gender gap in various contributions cannot be fully explained by population.*

Every figure in our article clearly indicates comparison with confidence intervals associated to a null hypothesis of no bias effects besides numerical differences between male and female populations. Therefore, our figures already showed the requested information.

Indeed, our statistical assessment – random shuffling of gender labels – leaves the network structure and the fraction of females at every level intact. In essence, we ask whether the observed contributions could have occurred randomly, if all actors of different gender behaved identically with respect to the contribution in question.

We apologize if this important procedural aspect was not clear enough. We have now better highlighted the concept of the shuffling method (cf. Figure 1) for the generation of gender-blind control networks, and edited the third paragraph of the Results section (third paragraph) in an attempt to make it more understandable. Furthermore, we have also improved figure captions.

*9) There are many interesting results that warrant more explanation in the text. At present, some of the most interesting results are buried in the figure captions. It would be great to see the authors embed some of these results, in more detail, in the main text.*

We thank the reviewer for this suggestion. Former Figure S3, now Figure 1, as well as former Figure S5, now Figure 3 are now presented in more detail in the Results section, (second and sixth paragraphs, respectively). We also better highlighted the homophily findings, which are a key novel contribution of our study. In particular, we have added a completely new section and figure panel about the observed different mechanisms of homophily for male and female editors (Figure 3) in the main manuscript.

We further rewrote and substantially extended the Discussion section of the manuscript to now include a discussion of these findings.

*10) Overall, the differences between observed and chance-level participation of women (for instance in Figure 2) are not that large, even though they are statistically significant. How concerned should we be about them in the first place?*

We agree with the reviewer that it is important to discern statistical significance, effect size, and relevance. First, while the differences, for the most part, seem small, they can reach up to a factor of 2 between the observed fraction of females with 10 contributions and the chance range. Given the large range in the number of occurrences, visualized in Figure 2, we had to use a logarithmic scale. This requires care when attempting to judge the magnitude of the bias just from the visual inspection of those graphs, while statistics are precise. Second, to highlight why our findings are concerning, we now include several sections in the revised Discussion that:

A) Argue why we are confident, in the light of previous findings, to state that discrimination is a critical reason for the remaining inequity (subsection “Evolution of participation rates by gender and causes for remaining inequity”, second paragraph);

B) List direct results of subtler forms of gender bias. Notably, even a slight homophily can influence and alter the way in which information spreads and is damaging to female scientists, whose work is perceived with a negative bias by the majority (subsection “Homophily in society and science”, last paragraph) and;

C) Translate the findings to specific structural disadvantages for women (subsection “Conclusions”, second paragraph).

*11) The authors advise that editors should be encouraged to make non-homophilic reviewer selection. But if the goal is to increase gender parity, shouldn't women editors be 'allowed' to be more homophilic, at least until the parity achieved? Might it be better to suggest that all editors should try to recruit more women reviewers?*

The reviewer is asking an interesting question and we agree: on first glance, it might seem like a prudent policy to support homophily among female editors to reach equity.

In response to question 1, we would like to highlight, however, that the goal of scientific policy is not equity per se, but equal opportunity and scientific meritocracy. To this end homophily should be reduced, too, as homophily, even in a situation when parity is reached, prevents equal opportunity. This is now detailed in the revised Discussion in the first two paragraphs of the subsection “Conclusions”.

In response to question 2, encouraging editors of both genders to recruit more women might foster the advance of excellent female scientists and advertise that women do good science. We stress however that a broad literature corpus in sociology of work environments have shown that the efficacy of top-down measures aiming at enforcing equal representation is poor if not paired with policies aiming at educating deciders about the importance of fighting gender discrimination (i.e. healing “from within” network construction mechanisms, rather than just artificially “transplanting” more female nodes into the network).

Considering both sides of the medal, it is unclear whether the positive effects (faster route to equity) or negative effects (undesirable behavioral trait) prevail in the long run. We added a corresponding paragraph to the Discussion (last paragraph) and the subsection “Conclusions” (third paragraph).